CANADIAN COLLEGE OF
HEALTH LEADERS
COLLÈGE CANADIEN DES
LEADERS EN SANTÉ

# Mexico: Lessons learned from the 2009 pandemic that help us fight COVID-19

2020, Vol. 33(4) 158-163
Health Leaders. All rights reserved.

journals.sagepub.com/home/hmf

Mauricio Hernández-Ávila, MD, MSc, DSc[1] ⓘ and
Celia M. Alpuche-Aranda, MD, PhD[2]

## Abstract

In April 2009, Mexican, American, and Canadian authorities announced a novel influenza that became the first pandemic of the century. We report on lessons learned in Mexico. The Mexican Pandemic Influenza Preparedness and Response Plan, developed and implemented since 2005, was a decisive element for the early response. Major lessons-learned were the need for flexible plans that consider different scenarios; the need to continuously strengthen routine surveillance programs and laboratory capacity and strengthen coordination between epidemiological departments, clinicians, and laboratories; maintain strategic stockpiles; establish a fund for public health emergencies; and collaboration among neighboring countries. Mexico responded with immediate reporting and transparency, implemented aggressive control measures and generous sharing of data and samples. Lessons learned induced changes leading to a better response to public health critical events.

## Introduction

In 2009, the Mexican Federal Government was prepared to detect and respond rapidly to the emerging threat from the A(H1N1)pdm09 influenza virus. Decisions were timely and resulted in effective actions to protect the Mexican population and to alert the international community of a novel influenza virus with pandemic potential.[1-3] By 2009, Mexico had advanced in planning and preparedness to mitigate the impact of a pandemic, with nine key milestones described in Table 1.

On April 14 and 15, 2009, the Ministry of Health (MOH) received notification of pneumonia cases affecting mostly young adults from Oaxaca, San Luis Potosí, and Mexico City. Samples from these cases were analyzed at the Laboratory for Epidemiological Surveillance of the Institute of Epidemiological Diagnosis and Reference (InDRE); however, either the result was negative or found to be positive for influenza "A" using Reverse Transcription Polymerase Chain Reaction (RT-PCR) but could not be further subtyped.

On April 17, the MOH issued an epidemiologic alert that mandated public hospitals to intensify surveillance for pneumonia. Data from public hospitals in Mexico City revealed that Intensive Care Units (ICUs) had an unusual number of pneumonia cases with a high proportion affecting young adults. Cases experienced a rapid clinical deterioration leading to severe, life-threatening viral pneumonia and high mortality.[5,6] In the same time period, the Centers for Disease Control and Prevention (CDC) in the United States (April 21) reported two influenza cases in southern California with gene segments that had not been reported previously among swine or human influenza viruses.

On April 22, following the protocols laid out in the Global Influenza Surveillance and Response System, Mexican samples were shared with CDC and the Public Agency of Canada. Samples were confirmed as influenza AH1N1pdm09, connecting the outbreaks occurring in Mexico and the United States.

On April 23, Mexican health authorities raised the alert level and gave the world an early warning. On April 24, the Federal Government mandated the implementation of various non-pharmaceutical interventions, which included social distancing measures such as self-quarantine for individuals experiencing mild symptoms, temporary school closure, and mass gathering (soccer games, concerts, church services, among others) postponements or cancellations. Within days, the MOH developed a mass media plan developed detailed within the Pandemic Influenza Response Plan (PIRP) and habilitated a national mass media campaign.[7] A post outbreak study reported that close to 90% of the Mexican population received a preventive message, which elicited some preventive behavioral change.[8] Protocols were released for the epidemiologic surveillance of sample collection, transport, and storage, along with medical attention including triage and clinical guidelines for treatment of patients. On-line courses for primary care physicians and hospital clinicians were also implemented. One lesson is that early involvement of the highest-level authorities of sectors involved in decision-making and pandemic response is pivotal. Mexico achieved this by convening an early meeting of the National Public Health Council (CSG for its acronym in Spanish), a body of the Mexican State that reports to the President with the authority to issue inter-sectoral mandatory interventions for pandemic control.

On April 26, CDC, United States, and National Medical Library, Canada, deployed staff to Mexico to support the outbreak response. The InDRE was reorganized according to

[1] Instituto Mexicano del Seguro Social, Dirección de Prestaciones Económicas y Sociales, Ciudad de México, México.
[2] National Institute of Public Health, Cuernavaca, Morelos, México.

**Corresponding author:**
Mauricio Hernández-Ávila, Instituto Mexicano del Seguro Social, Ciudad de México, México.
E-mail: mauricio.hernandeza@imss.gob.mx

**Table 1.** Summary of key milestones achieved by Mexico before de AH1N1 2009 pandemic and lessons learned

| Achievement | Lessons learned |
| --- | --- |
| Joining the international Global Action Plan to strengthen the public health response to the threat of international biological, chemical, and radionuclear terrorism (GHSAG) (2001). | International collaboration is fundamental to effective pandemic preparedness, response, and early interventions. |
| | Participation in this group was key to strengthen our emergency response capacities, as well as our collaborations with United States and Canada to assist each other and ensure a quick and coordinated response to pandemic emergencies. |
| Enacting the Law for the Creation of the National Committee for Health Security (2003). | Provided a legal mechanism to dictate measures at national level to control and investigate outbreaks. |
| Introducing vaccinations against influenza for children under 3 and adults over 60 years in the National Immunization Program (2003). | Provided protection against influenza and awareness regarding the importance of preventing seasonal influenza. |
| Publishing a Pandemic Influenza Response Plan (PIRP), based in a multi-sectoral operational strategy that planed the creation of a new influenza surveillance system (SISVEFLU), based on Influenza-Like Illness (ILI) detected in close to 300 primary care units around the country (2005). | The PIRP worked appropriately at Federal Government level. However local (state) governments lacked the organizational capacity to coordinate effectively with the plan. This created distrust and opposition to federal government non-pharmaceutical interventions (school closures and other). A continuous collaboration during the preparedness time is essential to improve the rollout of the PIRP in order to engage local governments and the full society in the response. |
| Establishing an Emergency Control Room (ECR) within the Ministry of Health (MOH) (2005). | The ECR played a key role in monitoring all ongoing activities. It also provided a space for international communications. Key lesson here was the need to develop this type of infrastructure at state level. |
| Testing and validating a full-scale exercise for pandemic response (2006). | The exercise failed to detect the complexity of local response. Our recommendation here is the need to involve local authorities. |
| | Table-top exercises may be a better solution, are less costly, and provide a better setting to understand the potential impact of a pandemic with the aim of facilitating appropriate contingency planning and preparedness actions. |
| Signing an agreement between the Mexican Government and Sanofi Pasteur to develop the local production of the influenza vaccine (2007).[4] | Pandemic preparedness collaboration among private businesses with the public sector is critically important. However, this mechanism failed to provide vaccines on time for the Mexican population. |
| Implementing diagnostic capabilities for influenza at 70% of the Public Health State Laboratories (PHSL) in the country. While the Federal Laboratory (InDRE) had the capability to perform detailed molecular subtyping of the influenza virus for routine surveillance. | Government should provide more resources and support for the development of real and effective surveillance systems. |
| A reserve of 1.3 million treatment courses of oseltamivir to cover 1.3% of the population. Broad-spectrum antibiotics used for treating bacteria super-infections, laboratory equipment, health personnel protection equipment, telecommunications equipment, and other supplies (2006). | The strategic stockpile was stored in bulk; reconstitution was difficult due to regulatory barriers. Other logistical problems surfaced, including shelf life and expiration of the new tablets, distrust of the public about its efficacy. |

Abbreviations: InDRE, Institute of Epidemiological Diagnosis and Reference; SISVEFLU: Epidemiological Surveillance System for Influenza; GHSAG: Global Health Security Advisory Group.

the PIRP, but within the first few weeks was quickly overwhelmed by more than 1,000 samples being received per day. Deployed US and Canadian experts helped to further define working areas and plans to expand diagnostics, and within 24 hours after their arrival, the testing line started at InDRE. The international collaboration was based on principles of respect and with the only goal to develop an appropriate laboratory response to this critical event.

## When did the circulation of the A(H1N1)pdm09 virus start in Mexico?

From January 1 to March 31, InDRE received 35,000 samples from around the country. The retrospective analysis of these

samples to detect AH1N1pdm09 detected 59 as positive for AH1N1pdm09. The earliest date was matched to a case from San Luis Potosi in north-central Mexico reporting symptoms onset on February 24, 2009; eight other cases from the same city occurred through March 31. These data support that the emergence of this new virus in Mexico may have occurred in the North and Central parts of the country 5-7 weeks before it was detected in Mexico City.

## Limitations in the implementation of PIRP

Initiating and maintaining an effective PIRP for influenza depended on multiple health system features at the time, of which we highlight the following:

1. Developing an effective surveillance and diagnostics system: An effective notifiable surveillance system is among the most critical elements necessary for outbreak early detection and for implementing a timely and adequate response. Before the outbreak, Public Health State Laboratories (PHSL) facilities were under-funded and under-staffed, with only a few having immunofluorescence diagnostic capabilities for influenza.

2. Health system fragmentation: The local governments have the responsibility to channel federal financial support for supplies, equipment, and personnel for the PHSL network. Influenza laboratory confirmation was centralized at InDRE, including most of the subtyping and viral isolation. Although legal provisions were in place for PHSL to notify the MOH regarding influenza cases, this was not occurring. For example, for the year 2008, the National Surveillance System Weekly Report totaled 23.3 million cases of acute respiratory diseases and 138,839 cases of pneumonia, with only 151 cases diagnosed as influenza; this last number reflects a substantial underreporting.

3. Information system maturity: Similarly, the epidemiological surveillance system for influenza (SISVEFLU) was reporting at 23% of its planned capacity, and information derived from this system was operated using an ad hoc paper-based surveillance system and provided limited information regarding Influenza-Like Illness (ILI)/SARI, with under-representation of hospitalized severe acute respiratory infections cases.

4. Responsible science communication and media: Diagnosis at PHSL facilities relied in immunofluorescence which are known to have a low negative predictive value. This created political misunderstanding and distrust, as some politically sensitive samples such as those reported as negative by states with no AH1N1pdm09 cases were later confirmed as positive by the federal laboratory, using Quantitative RT-PCR (qRT-PCR) test. The media quickly caught on to this discrepancy and created unfavorable media coverage to the response.

5. Healthcare provider planning and preparedness: With regard to medical care and the organization of ICUs, the medical staff in many hospitals were not adequately prepared for prompt recognition and treatment of serious forms of influenza in young adults. There was a significant lack of planning for hospital reorganization, triage, and treatment of an excess caseload of influenza. Intensive care units did not have PIRP in place to develop capacities for the epidemic, including but not limited to insufficient personnel, supplies, and equipment for mechanical ventilation. These limitations and the fact that Mexico was one of the first countries affected may explain the high mortality observed in comparison with other countries.[9,10]

6. Weak regulatory and logistic frameworks: The strategic stockpile of oseltamivir was stored in bulk powder containers, with the PIRP stipulating that reconstitution would be carried out at InDRE and the PHSL network. However, it immediately became apparent that it had been a mistaken assumption. These laboratories had neither the facilities, training, nor the legal authorization to fill prescriptions or reconstitute the powder into tablets or liquid for direct prescription. This plan ended in a complex regulatory issue and delayed the use of the stockpile. In the end, tablets were fabricated with the collaboration of the original pharmaceutical provider. Other logistical problems surfaced, including shelf life and expiration of the new tablets, distrust of the public about its efficacy, and others that the PIRP did not consider.

7. Lack of emergency funds: Another limiting factor was the lack of an emergency response fund or norms to conduct expedited purchasing procedures. This restricted the provision of needed supplies or equipment (aside from antivirals and antibiotics, and the laboratory network, which did receive funding). For example, the lack of funding limited the conduction of the initial field investigations needed for interviewing and collecting blood samples from patients diagnosed with or exposed to AH1N1pdm09. This is an important issue for future events, as a rapidly available mechanism is needed to support outbreak control activities. Funding mechanisms are needed to support activities during and after a crisis; the ideal circumstance is a continuous and sustainable source to support a stronger response.

8. Lack of state and local engagement in PIRP development: One opportunity for improvement identified was the lack of operational pandemic response plans at the state government level; states lacked stockpiles of supplies and medical drugs, as well as extended diagnostic capabilities for adequate detection and surveillance. The PIRP plan was developed for the federal level, with very little participation from state governments.

## Actions taken

1. Developing an effective surveillance system: An effective notifiable surveillance system was found to be among the most critical elements necessary for outbreak early detection and for implementing a timely and adequate response.

2. Access to emergency funding: Funding released during the outbreak was key to support the surveillance network. The InDRE received funds for the acquisition of qRT-PCR equipment, biosafety cabinets, DNA/RNA extraction robots, and freezers. Within the first 4 months of the outbreak, the entire country, through the PHSL network and some larger hospitals, had staff, reagents, and equipment to diagnose the new virus. This dramatically improved Mexico's influenza surveillance system which was transformed into an integrated epidemiologic and virologic national network.

3. Strengthening information systems: As a response, Mexico expanded its monitoring capabilities with what are now 712 Influenza Monitoring Health Units for ILI and SARI. These units help centralize data and are easily integrated within a digital platform that allows data transfer in real time between members of the SISVEFLU network.

Characteristics of the SISVEFLU integrated network:

   a) Strong qRT-PCR-based virologic surveillance.
   b) Sequencing analysis for detailed pathogen characterization.
   c) Continued quality assurance through both the CDC's influenza division and the Hong Kong regulatory laboratory.

4. Access to life-saving medication: Though shortages did appear worldwide, the problem was mitigated by donations of more than 700,000 treatments from Roche Laboratories, the World Health Organization (WHO), and the governments of the United States and France immediately after declaration of the epidemic in April 2009. In addition, the MOH bought 900,000 additional treatment courses of oseltamivir and 100,000 of zanamivir.

## Lessons learned

The H1N1pdm09 pandemic signaled a new era of global epidemics and the effect of globalization. At a higher level, one of the key lessons learned and felt worldwide is that outbreak containment in modern times seems unlikely. The experience in Mexico calls to attention the following:

*Containment strategies might have limited success:* The novel AH1N1pdm09 pandemic virus moved within and beyond Mexico too quickly for social distancing policies to impact in full force and spread rapidly to other regions of the world. Alerts discouraging non-essential travel to Mexico served no purpose as they did not contain the outbreak and did not prevent its further international spread. By the end of the year, more than 100 countries were reporting cases. This observation is consistent with the ample geographical spread of other emergent infections such as Severe Acute Respiratory Syndrome (SARS; 29 countries) and Middle East Respiratory Syndrome (27 countries), and currently, SARS-CoV-2 started in China and currently is quickly spreading across the globe to 58 other countries.

*Early warning, response and preparedness are a must*: CDC estimates that 105,700-395,600 people worldwide died from a cause associated with AH1N1pdm09 influenza; 80% of deaths are estimated to have occurred in people younger than 65 years of age.[11-13] In order to prevent global spread of the next influenza pandemic, public health officials need not only to predict the location but also to detect it within weeks of its emergence. New methods that rely on artificial intelligence may at some point in time increase our ability to predict and detect earlier these threats. At the time, both goals seem impossible, as there will always be a delay between the emergence of the outbreak's index case and the detection of the outbreak by healthcare providers, laboratories, or public

health authorities. The goal is to minimize this delay.[2] However, early warning was key for vaccine development and for other countries to prepare.

*Early cases are the most vulnerable*: When a new infectious agent causes an outbreak, the first country affected most likely suffers the most. Mexico experienced a higher AH1N1pdm09 influenza mortality burden than other countries for which estimates are available; of note was the particularly high case fatality ratio reported (the case-fatality ratio among ILI cases was 1.2% overall and 5% among laboratory-confirmed A/H1N1, compared to 0.05% and 0.03% in the United States and United Kingdom, respectively).[5,6,14-16] It is estimated that in the first year after the emergence of the (H1N1)pdm09, between 28% and 34% of the Mexican population was infected with the new virus,[4] disproportionately affecting individuals aged 5-59 years. According to Charu et al., the (H1N1)pdm09 was associated with 445,000 Years of life lost (per 100,000 population) between April and December 2009[11] and a mortality burden 0.6-2.6 times that of a typical influenza season.[12]

*Enhance early detection and contact tracing:* The fact that the pandemic began in the area around the Mexican-US border or Northern central part of Mexico, rather than, for example, in South East Asia, led to another important lesson: influenza pandemics can start in any location.[17] The original pandemic plan assumed a scenario for protecting Mexico from an outbreak originating in East Asia, with a 6- to 10-week delay before the detection of first cases. Looking back at the early response, a rapid containment strategy is crucial to control or reduce the pandemic risk, with movement restriction, antiviral prophylaxis, and public health interventions to reduce social contact. Detection, investigation, and reporting of the first cases, with appropriate contact tracing, must happen quickly. The first weeks of an outbreak are critical for developing a thorough understanding of transmission dynamics and severity of the disease. Decisions at this stage need to be made quickly, but unfortunately, decision are made under a considerable degree of scientific uncertainty.

*Prepare for the economic impact:* It is also important to consider that public health emergencies related to infectious diseases evolve rapidly and are costly events for the world economy and even more costly for the countries that are affected first. In economic terms, however, the 2009 pandemic in Mexico had an estimated impact of about 0.7% of gross domestic product.[4,18]

*Data timeliness shapes the appropriateness of the response:* In retrospect, we now know that the virus showed moderate virulence. Decisions were made when the new virus had not yet been fully identified and little was known about its clinical course. Furthermore, the real epidemiologic situation at the population level was masked by the lack of critical information from our surveillance system and by reports from hospital emergency departments.

*Prepare at all levels:* While the Mexican PIRP proved incredibly useful, for countries like Mexico, where states have their own health laws, the evaluation and real development of local PIRPs should be promoted.

*Arrange funding in advance:* Our recommendation is that PIRP should establish a public health emergency fund of at least US$10 million that would be readily accessible during an outbreak. The precise conditions and terms for use of the fund should be preapproved.

*Seek greater international collaboration and equitable access to vaccines:* Affected countries shared virus samples with WHO for surveillance and vaccine development without asking any benefit. Early sharing of samples and international collaboration made possible the development, testing, and production of a targeted AH1N1pdm09 influenza pandemic vaccine within 3 months. However, not enough of the vaccine was produced to meet the demand; it arrived at a high price (2-3-fold the price of the seasonal vaccine), and countries negotiated individually in a closed format. Equitable access to vaccines was not achievable; Mexico received its first vaccine shipment late in December 2009, several months after Canada and the United States had started their vaccination programs. The lesson here is at global level; countries need to work in a new framework to improve equitable access to influenza vaccines in pandemic situations. The efforts to establish national self-sufficiency for influenza vaccine were not ready at the time of the outbreak. This effort is encouraging but until now has failed to guarantee this important health security issue[19] and need attention.

*Improve international regulatory frameworks:* We propose that cost of vaccine production in the context of a global pandemic should be public knowledge so that a fair price can be reached. Markets could be controlled to prevent unfair trade and fix profits in advance. When a global pandemic emerges, we need to have mechanism in place to address it with equity and solidarity.

*Develop continuous education programs.* Other key lessons learned from the AH1N1pdm09 influenza outbreak of 2009 include the need to continually reinforce infection control education and communication, vaccination of health personnel, along with the use of personal protective equipment. In Mexico, tertiary care facilities reported large numbers of health workers falling sick because of the inappropriate use or lack of protective equipment.

*Global problems require global solutions:* For Mexico, international collaboration was a core element to have pre-established technical procedures with harmonized international standards and to interchange experiences and build capacities to achieve accurate and appropriate responses to critical public health events. Joint international activities based on mutual respect, with the objectives of technical competence and ensuring the safety of the population and surrounding regions, will always reinforce and improve the capacity to respond to critical public health events, creating the same opportunities worldwide.

## Conclusion

The Mexican PIRP, including all preliminary preparation, was a supportive and decisive element for Mexico's response to the influenza pandemic in 2009. Mexico's surveillance systems captured hospitalization, case fatality, and mortality impact in near real time. Decisions were made under conditions of uncertainty but were effective to protect the Mexican population, and our country alerted the international community about a new novel influenza virus with pandemic potential, transparently and on time.

Policies implemented were considered by some as excessive, but it is said that "no one is a prophet in his own land." Constructive criticism should be based only on what was known at the time of decision-making and not on what was learned subsequently. Despite the limitations of the Mexican PIRP previously described, it was instrumental for the early response. In the words of Margaret Chan,[3] WHO Director General in the period, "Mexico gave the world an early warning, and it also gave the world a model of rapid and transparent reporting, aggressive control measures, and generous sharing of data and samples."

Lessons learned propelled important changes in Mexico's preparedness and infrastructure for pandemic response. Organizations have advanced InDRE and state laboratories have been reinforced, and SISVEFLU is now working at full capacity. Mexico is still working toward self-sufficiency in vaccine production, and contracts with pharmaceutical industries are being negotiated. We strongly believe that if we continue to share what we have learned, a more secure and healthier world will emerge.

Several interventions designed and deployed out of the H1N1 response have proved resilient and are now strong components of the systemic response to COVID-19.

Firstly, improvements in the surveillance system that included the deployment of the Sentinel Surveillance Program, allowed the swift addition of a COVID-19 module. This new element of the program helped identify possible COVID-19 cases even during early phases of the COVID pandemic in Mexico. The amended program has allowed health authorities to plan scaled social distancing interventions and begin contact tracing early on.

Likewise, the laboratory infrastructure, strengthened in capacity and scale during H1N1, has made it possible to deploy an early implementation of SARS-CoV-2 Rt-PCR diagnosis throughout the country. The Public Health Laboratory Network proved as a useful scaffolding that enhanced the laboratory system by leveraging and incorporating support from private and public laboratories and hospitals.

Lastly, capacity building has been very successful. As a part of an ongoing educational platform, a new training program for laboratory professionals began in February 2020. This training has improved the capacity for Public Health Laboratories to develop COVID-19 molecular diagnostic of COVID-19. Aimed at a larger, more general audience, on-line capacity building to general audiences (over 1 million people) has proven very popular and has helped deliver best practices to help the public identify alert signs and symptoms, indications on self-isolation, and other guidance on behavior aimed to reduce transmission rates at home and in the community.

On the other hand, as the Mexican health system responds to COVID-19 pandemic, we identify areas where lessons learned from H1N1 were not applied as readily as they should have. We mentioned that early involvement of the highest-level authorities was key to an adequate and comprehensive response to an epidemic, yet the CSG did not convene an extraordinary session before SARS-CoV-2 arrived in Mexico, which delayed inter-sectoral preparedness.

Furthermore, the H1N1 may have given us a false sense of security in regard to the global market for protective equipment, laboratory resources and access to medical supplies. In 2009, Mexico saw the very first cases of H1N1, and the global support was immediate. The situation could not have been more different during for the COVID-19 pandemic, these same resources are now extraordinarily difficult to secure. Wealthy countries where SARS-CoV-2 took an earlier toll in 2020, are consuming a high proportion of the global demand for medical supplies. This situation has distorted global markets and has negatively affected Mexico's testing strategy, reduced PPE availability, and contribute to scarcity in medication and life-saving biomedical equipment like ventilators.

## Declaration of conflicting interests

Dr. Hernandez-Avila was Subsecretary of Health during the AH1N1pdm09 and coordinated control actions. Dr. Alpuche-Aranda was head of the Institute of Epidemiological Diagnosis and Reference (InDRE) at the time of the outbreak.

## ORCID iD

Mauricio Hernández-Ávila https://orcid.org/0000-0002-0393-7590

## Supplemental material

Supplemental material for this article is available on-line.

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
