## [Supplemental Material, Reviewer_comments - Mexico: Lessons learned from the 2009
pandemic that help us fight COVID-19 · Healthcare Management Forum]

Reviewer comments	Author response
Is the topic relevant and timely to health leadership?	
Yes, certainly with the current COVID-19 situation, this article is very timely. The policy and public health recommendations are clear and actionable.	We thank the reviewers for their comments
Yes. This topic is relevant and timely to health leadership, as leaders at every level of healthcare need to improve their awareness of, and preparedness for, diseases with pandemic potential.	
How original is the work?	
This paper brings together a summary of "lessons learned" from credible source materials related to this pandemic influenza. Original analysis and recommendations are presented.	We thank the reviewers for their comments
I find it original, as it gives a view of an outbreak that attracted significant international attention from what appears to be a vantage view of its epicenter.	
Does the content advance knowledge in the practice of health leadership?	
This is not my area of expertise; however, the paper appears to contribute to knowledge of best practices in the areas of public	We thank the reviewers for their comments

health management and health policy related to pandemic planning and response.	
Yes. It enhances to the conversation around what could work, or not, in preparation for and responses to infectious diseases with pandemic potential.	
How well is the connection made between theory and practice?	
Practice recommendations are drawn directly from this case study. There is little reference to public health theory or existing best practices to provide context for the paper's recommendations.	We have expanded the discussion section to address this issue, the word extension form the editorial board makes this a challenge.
It was made well. Distinctions were made between what standard practices (theory) should have been, and what the limitations were to their actual application (practice).	
Do the authors use sound methodology?	
The paper does not include a methodology section. The approach for identifying the relevant literature (published and grey) was not specified. And, the method(s) for drawing themes from the reference materials was not discussed.	We have clarified our roles in the response to AH1N1 in Mexico which is linked to the information presented.
It appears to be a straight-forward and informative accounting of the early response to an outbreak.	The authors played different roles during the response to AH1N1 in Mexico. Dr- Hernandez-Avila was undersecretary of Health and Dr- Alpuche Aranda was head of INDRE.
Are the conclusions appropriate and are they generally applicable to other situations?	

It is unclear to me how the authors synthesized and developed their conclusions -- therefore it is difficult to judge if they are appropriate and/or fulsome. For instance, in reviewing one referenced article (https://health-policy-systems.biomedcentral.com/articles/10.1186/1478-4505-7-21#Sec12), I note that mental health supports were identified as important -- but this was not included in the lessons learned within this paper.	We think is impossible to address all areas of a complex response. We agree with the reviewer and we have addressed this issue in the manuscript.
Yes.	
What can be done to improve this paper that has not been specified above?	
The Table 1 Summary of key milestones achieved by Mexico before the AH1N1 2009 pandemic would be complemented by having a summary of the lessons learned from the pandemic response. Also, separating the public health management lessons from the health policy / political lessons would be useful.	Table 1 has been modified accordingly
1. In the introduction: response to this outbreak was said to be timely and effective. What was this relative to? It is essential to differentiate it from the opinion of the authors.	We have added the proper references.
2. The introduction subsequently goes into a detailed timeline of events around the outbreak. A summary of these events would be more appropriate for the introduction, and a separate timeline section preferable for showing these trends after the introduction. The timeline view used for the pre-outbreak activities at the end of	

the paper should be an effective method for the outbreak timeline as well.	
3. Itemizing the points in the "Limitations in the Implementation of Pandemic Influenza Response Plan" with sub-sections/sub-headings, should make them easier to identify and contextualize.	We have made the suggested changes
4. Some actions and outcomes (e.g., regarding funding released during the outbreak) were mixed into the "Limitations' section, and would be better in their own section to draw appropriate attention to that content. This might be better before the "Lessons Learned" section.	We have made the suggested changes
5. Itemizing lessons learned (not in paragraph structure) might improve reading flow, and identification of disparate points.	We have made the suggested changes
6. Some limitations (e.g., lack of an emergency response fund") were captured in the "Lessons Learned" section. These can be moved to the "Limitations" section, whatever lessons were learned from them left in the "Lessons Learned" section.	We have made the suggested changes